# *SilentStriker*: Toward Stealthy Bit-Flip Attacks on Large Language Models

**Haotian Xu[1]**   **Qingsong Peng[1]**   **Jie Shi[2]**   **Huadi Zheng[2]**   **Yu Li[1]***   **Cheng Zhuo[1]**

[1] Zhejiang University       [2] Huawei

## Abstract

The rapid adoption of large language models (LLMs) in critical domains has spurred extensive research into their security issues. While input manipulation attacks (*e.g.*, prompt injection) have been well-studied, *Bit-Flip Attacks (BFAs)*—which exploit hardware vulnerabilities to corrupt model parameters and cause severe performance degradation—have received far less attention. Existing BFA methods suffer from key limitations: they fail to balance performance degradation and output naturalness, making them prone to discovery. In this paper, we introduce *SilentStriker*, the first stealthy bit-flip attack against LLMs that effectively degrades task performance while maintaining output naturalness. Our core contribution lies in addressing the challenge of designing effective loss functions for LLMs with variable output length and the vast output space. Unlike prior approaches that rely on output perplexity for attack loss formulation, which in-evidently degrade the output naturalness, we reformulate the attack objective by leveraging key output tokens as targets for suppression, enabling effective joint optimization of attack effectiveness and stealthiness. Additionally, we employ an iterative, progressive search strategy to maximize attack efficacy. Experiments show that SilentStriker significantly outperforms existing baselines, achieving successful attacks without compromising the naturalness of generated text.[1]

## 1   Introduction

Large Language Models (LLMs), equipped with the capacity of text understanding, reasoning, and generation tasks, have been widely adopted in critical domains such as economic systems, social services, and healthcare [1, 2, 3]. As their adoption accelerates, the need for rigorous assessment of their safety and reliability has become more pressing [4, 5]. While input-based attacks have been extensively studied, hardware-level threats—such as Bit-Flip Attacks (BFAs)—remain underexplored. BFAs exploit low-level vulnerabilities like RowHammer to induce bit flips in DRAM, potentially corrupting memory regions that store model weights and compromising model integrity [6].

Several studies have begun to explore the vulnerability of LLMs under BFAs. PrisonBreak [7] reveals that targeted bit flips (fewer than 25) can bypass safety mechanisms in LLMs and induce harmful behaviors, though its attack scope is limited to specific functionalities, leaving overall model performance largely intact. Similarly, GenBFA [8] demonstrates that as few as three bit flips can severely degrade performance, but the generated outputs often become incoherent or nonsensical, making these attacks easily detectable [9].

To address the above challenge, we propose a stealthy and effective bit-flip attack against LLMs, named **SilentStriker** (illustrated in Figure 1). However, achieving both stealthiness and effectiveness

---

*Corresponding author. Email: `yu.li.sallylee@gmail.com`

[1]Code is available at: `https://github.com/HaotianXu1/SilentStriker`

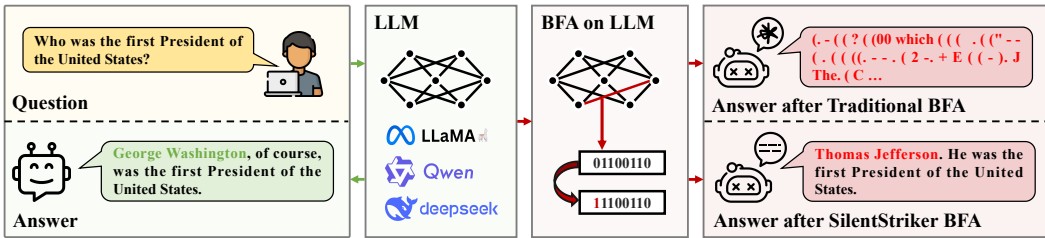

Figure 1: **The goal of our *SilentStriker*.** Unlike previous methods, our method can compromise the model outputs in a stealthy manner.

in LLM-targeted attacks presents unique difficulties. Unlike in CNN-based attacks—where the target class can be explicitly leveraged to guide the attack loss—LLMs produce variable-length outputs with inherently uncertain content, rendering direct use of generated text in loss formulation infeasible. Alternatively, attacking via output perplexity (*e.g.*, increasing perplexity to degrade answer quality) often leads to unnatural outputs, which makes such attacks easily detectable. To overcome this limitation, our key insight is to construct the attack loss based on the suppression of critical output tokens, effectively preventing them from appearing in the attacked model's generation, while maintaining overall output perplexity within natural bounds. This approach enables SilentStriker to significantly degrade task performance while preserving output fluency and coherence, thereby enhancing both the stealth and impact of the attack in real-world scenarios.

We summarize our contributions as follows:

- To the best of our knowledge, we are the first to propose a stealthy bit-flip attack targeting LLMs. By flipping just a few bits (e.g., around 50 bits for our evaluated settings) out of billions of parameters, this attack significantly degrades the performance of LLMs while remaining difficult to detect, even for quantized models.

- We propose a token-based method to tackle the intrinsic challenge in balancing the model performance degradation and the output naturalness objectives. The core at this approach is a differentiable token-based loss for degrading model performance, with which we can leverage the perplexity for improving the naturalness. Furthermore, to enhance output fluency and improve attack efficiency, we enhance the token-based loss with the key-token based loss.

- We propose an iterative and progressive search strategy to identify the optimal layer and location for the attack, minimizing the number of bits required for successful stealthy attacks. Moreover, for FP4-quantized models, we propose an improved bit selection strategy to enhance attack efficiency.

We validate our approach through extensive experiments on multiple popular LLMs and tasks, demonstrating significant task performance degradation and output naturalness compared with baselines. For example, in LLaMA-3.1-8B-Instruct INT8-quantized model, after flipping 50 bits, accuracy on GSM8K dropped from 65.7% to 7.6% while the naturalness score evaluated by GPT of the output dropped only from 66.0 to 61.1. Compared to GenBFA, although the accuracy dropped to 0%, it causes a complete collapse in output fluency, with the naturalness score dropping to 0 and perplexity skyrocketing to $5.5 \times 10^5$.

## 2    Related Work

**Large Language Models (LLMs).** LLMs already exhibit strong capabilities in natural language dialogue, text creation, and information analysis, with their scope now extending to more sensitive domains such as medical diagnostics, legal document analysis, and policy consultation [1, 2, 3]. Their fundamental principle lies in training DNNs on massive text corpora, iteratively refining network parameters to learn the statistical distributions and structural patterns of language [10]. Typically composed of layers such as embedding layers, attention layers and Multilayer Perceptron (MLP) layer, these layers work in tandem to capture intricate linguistic and semantic dependencies [11]. The

core of their text generation process lies in predicting the probability distribution for the next word or token: at each step, the model considers the existing contextual input to produce a distribution over candidate words and either samples or selects the most probable token. By leveraging the linguistic and semantic knowledge learned from extensive data, LLMs can automatically generate natural language text that is both coherent and readable.

**BFA against DNNs.** BFAs are hardware-level adversarial techniques that manipulate neural network parameters by intentionally flipping bits in memory, thereby corrupting model behavior [12]. These attacks typically leverage disturbance errors in DRAM [13], such as those induced by Rowhammer, where repeated memory row accesses cause charge leakage in adjacent rows, leading to unintended bit-flips [14]. In general, existing BFA against DNNs can be categorized into untargeted and targeted attacks. Untargeted attacks aim to degrade the overall predictive accuracy of the model across all inputs, potentially reducing its performance to the level of random guessing. To this end, adversaries flip the bits with the highest gradient of the inference loss, leading to substantial prediction errors [12, 15]. In contrast, targeted attack seek to manipulate the model's output for specific inputs, enabling more stealthy attacks. However, due to the need to maintain the accuracy of the non-target inputs, locating vulnerable parameters becomes more complex than in untargeted attack. To address this challenge, recent works craft loss functions to identify class-sensitive parameters. Some methods focus on maximizing the output probability of the target class or the confidence gap between the target and original classes to induce targeted misclassification [16, 17]. Others further incorporate constraints on parameter perturbation and preserve the original predictions on non-target inputs, thereby improving stealth and reducing unintended effects [18].

**BFA against LLMs.** In language models, BFAs pose unique risks due to autoregressive generation; a single corrupted weight can cascade errors across tokens [19], enabling unintended responses with minimal footprint. Recently, PrisonBreak [7] and GenBFA [8] have successfully extended BFA to LLMs with billions of parameter scales. PrisonBreak designs a BFA methodology specifically for jailbreaking aligned LLMs. By combining gradient-guided BitFinder, the attack flips 5–25 bits in billion-parameter models to disable safety mechanisms. BitFinder employs a progressive search to iteratively flip bits that maximize harmful response likelihood while minimizing utility loss, prioritizing exponent bits in half-precision weights. The results of their experiments show that flipping just 3 bits in LLaMA-2-7B reduces refusal rates by 86% while retaining 98% of benign task accuracy. This work reveals that alignment techniques like RLHF are brittle to precise parameter perturbations, as even highly secure systems lack safeguards against targeted bit-level corruption. AttentionBreaker challenges the presumed robustness of transform-based LLMs to BFAs. Using GenBFA, an evolutionary algorithm, the framework identifies sparse critical weights through a hybrid sensitivity metric and optimizes bit selections via genetic operations. In LLaMA-3-8B, perturbing 3 bits ($4\times10^{-9}$% of total parameters) reduces MMLU accuracy from 67% to 0% and increases perplexity from 12.6 to $4.7\times10^5$. This highlights a paradox: while LLMs' scale suggests redundancy, their reliance on sparse critical parameters amplifies vulnerability to minimal adversarial perturbations.

In the above BFAs to LLMs, PrisonBreak aims to bypass safety mechanisms by flipping a few bits to trigger harmful outputs, but it does not impact the model's performance in typical scenarios. GenBFA targets key weights to degrade performance; however, this leads to a significant increase in perplexity, causing the model to generate gibberish outputs and making the attack easily detectable. In contrast, our research focuses on degrading performance without significantly raising perplexity, providing a stealthy BFA approach to LLMs. Our goal is to produce answers that look fluent and coherent but are wrong, which makes detection harder in practice and increases the risk that users trust and act on them.

## 3 Methodology

### 3.1 Threat Model

We consider a threat scenario where the attacker's target is LLM deployed on edge devices, lacking server-grade hardware protections such as error-correcting code (ECC) memory. Given the high cost and typical server/workstation confinement of ECC hardware [20], this deployment context is realistic for consumer laptops, desktops, and mobile platforms.

Following prior BFA work [7, 8, 12, 15], the adversary is white-box: they know the model architecture, parameters, and the memory layout—i.e., where weights reside in DRAM—and possess RowHammer

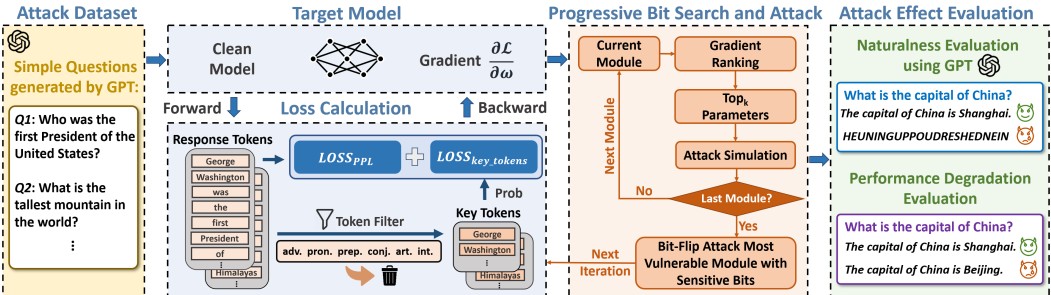

Figure 2: The overview of our *SilentStriker* framework. During loss calculation, the target model is forward-propagated on the attack dataset to generate responses. Perplexity loss is computed over the full token sequence, after which responses are filtered to retain key tokens; their aggregated probabilities define a key-token loss. The overall loss is the sum of these two components, and back-propagating this loss yields gradients for all parameters. In the Progressive Bit Search phase, independent simulated attacks are conducted on each LLM module to identify the most vulnerable one. Within each module, parameters are ranked by gradient magnitude, and the top-k parameters with the largest gradients are selected.

capabilities. The attacker's objective is to degrade the model's ability to answer correctly while preserving output naturalness and coherence, reducing the chance of detection during ordinary use. The attacker operates with only standard user-level privileges, obtained by distributing an ostensibly benign application (e.g., a utility, game, or productivity tool) through common channels. When executed by an unsuspecting user, the payload runs under user permissions which, while short of kernel control, suffice to interact with user-space process memory.

Operationally, we assume that once the model's weights are loaded, their addresses remain stable for the lifetime of the inference process. Although DRAM is periodically refreshed, refresh cycles read and immediately rewrite the same cells without relocating data [21], and address remapping does not occur during normal process execution [22]. Under these conditions, targeted disturbance-induced bit flips can be planned and executed reliably against static weight locations.

## 3.2 Attack Framework

Our SilentStriker framework is shown in Figure 2. At first, we construct a simple attack dataset, which is generated by GPT-4o. We evaluate the victim model on this dataset using a composite attack loss, composed of Key Tokens Loss and Perplexity Loss. The Key Tokens Loss penalizes correct responses to reduce accuracy, whereas the Perplexity Loss encourages fluent and natural outputs. After backpropagating the loss to obtain gradients, we use gradient-based ranking in the Progressive Bit Search stage to progressively identify vulnerable modules for targeted bit-flip attacks. Finally, we assess the naturalness of the model's outputs by combining perplexity (PPL) with evaluations from a GPT-based judge.

## 3.3 Attack Loss Calculation

The attack objectives can be accomplished by carefully designing the loss function. One of our objectives is to degrade model performance, which is typically achieved by increasing the Cross Entropy loss [8]. Another equally important objective is to maintain stealthiness, which entails preserving the naturalness of the output. This can be achieved by minimizing Perplexity. Perplexity reflects the model's confidence in predicting the next token given its context; a lower value typically indicates more fluent and coherent outputs, closely resembling human language patterns [23]. However, increasing Cross Entropy inevitably leads to higher Perplexity, as the latter is the exponential of the former, making their objectives inherently contradictory when used together in the loss function. Consequently, identifying an alternative loss function component to distinctly direct model performance reduction becomes essential.

A potential solution to replace cross entropy is semantic similarity. Nonetheless, computing semantic similarity usually necessitates a embedding model, such as Sentence-Bert [24], which involve non-differentiable tokenize and detokenize processes. Therefore, using semantic similarity as a loss

function presents significant challenges. Hence, our core challenge is to design a loss function capable of effectively reducing model performance without conflicting with perplexity and differentiable.

It is well known that large language models generate text token by token [11]. At each step, the model outputs a probability distribution over the entire vocabulary and then samples from it to produce the next token [25]. This means that for every output step, we have access to the predicted probability of any token. Since these probabilities are derived by applying a softmax operation to the output logits, they are fully differentiable. By lowering the probability assigned to the original output token, the model is encouraged to generate alternative tokens, thereby deviating from its original behavior and degrading its performance.

Based on this insight, we propose a token-based loss that explicitly guides model performance degradation by leveraging the prediction probabilities of the original tokens in the output.

**Key Tokens Loss.** However, computing the token-based loss using the probabilities of all tokens in the original output introduces several issues. In a typical model-generated sentence, many tokens that mainly contribute to sentence fluency and cohesion, such as conjunctions, prepositions, carry little semantic content and are often unrelated to the input question. Including these tokens in the loss calculation increases computational cost and memory usage without providing additional useful information. Moreover, these functional words play an essential role in maintaining the fluency and coherence of the sentence. Suppressing their probabilities can disrupt the sentence structure and contradict our goal of preserving output naturalness. To address this, we further propose the Key Tokens Loss, which only considers the probabilities of key tokens—defined as the remaining tokens after removing all adverbs, pronouns, prepositions, conjunctions, articles, interjections, and punctuation.

To compute Key Tokens Loss, we first extract the key tokens. Through a forward pass of the victim model on the attack dataset, we obtain its responses. Then, we load a pre-trained spaCy model (en_core_web_sm) [26] to tokenize and assign part-of-speech tags to each response, filtering out the previously defined non-key categories. The remaining words become our key words, which we then feed into the LLM's tokenizer to produce the final key tokens.

For example, given the original prompt '**What is the tallest mountain in the world?**', the original response is '**Mount Everest, located in the Himalayas on the border between Nepal and Tibet...**', and after removing the non-keyword components, the remaining '**Everest**', '**Nepal**', '**Himalayas**', '**Mount**', '**Tibet**', '**border**', '**located**' represents the keywords. By tokenizing the above words, we obtain the key tokens set $\mathcal{K}$.

Logits represent the raw scores that the LLM assigns to each word or token; the higher the logits, the greater the probability of that token being output. Through softmax normalization, we convert the logits into probabilities, obtaining the output probability for each token. To suppress the model's probability of outputting the correct token, we sum the probabilities for the key tokens and use the square of the sum as the accuracy suppression part of the loss function, which non-linearly amplifies the penalty for larger sums, thereby providing stronger gradients to more effectively suppress high probabilities. As shown in Eq.(1), $L_{\text{key\_tokens}}(x, \mathcal{K}; \theta)$ denotes the key tokens loss, where $x$ is the input sequence, $\mathcal{K}$ is the set of key tokens, $\theta$ represents the model parameters, $N$ is the total number of output tokens, and $p_\theta(t \mid x, i)$ is the probability the model assigns to token $t$ at position $i$ given $x$; the double summation therefore accumulates these probabilities over every key token $t \in \mathcal{K}$ at each position $i$ and across all positions from 1 to $N$.

$$L_{\text{key\_tokens}}(x, \mathcal{K}; \theta) = \left( \sum_{i=1}^{N} \sum_{t \in \mathcal{K}} p_\theta(t \mid x, i) \right)^2 \tag{1}$$

**Perplexity Loss.** To maintain the naturalness of the response and the stealthiness of the attack, we need to reduce perplexity while implementing the attack, therefore, a penalty will be applied to any increase in perplexity. As shown in Eq.(2), $L_{\text{PPL}}(x; \theta)$ denotes the Perplexity Loss, and $p_\theta(y_i \mid x)$ is the probability the model assigns to token $y_i$ given $x$. The outer exponential simply converts the negative average log-likelihood back from log space into the perplexity metric. In contrast to previous work [8] that generally included a PPL term aimed at increasing perplexity (*i.e.*, minimizing $-L_{\text{PPL}}$), we take the opposite approach and employ $L_{\text{PPL}}$ directly without a negative sign.

$$L_{\text{PPL}}(x; \theta) = \exp \left( -\frac{1}{N} \sum_{i=1}^{N} \log p_\theta(y_i \mid x) \right) \tag{2}$$

**Final attack Loss.** As shown in Eq.(3), final attack loss equals to the sum of Key Tokens Loss and Perplexity Loss.

$$L_{\text{attack}} = L_{\text{key\_tokens}}(x, \mathcal{K}; \theta) + L_{\text{PPL}}(x; \theta) \tag{3}$$

### 3.4 Progressive Bit Search

After computing the loss function and backpropagating to obtain the gradients, we proceed to the Progressive Bit Search phase. In this phase, we need to undergo independent simulation attacks for each module in LLM to identify the most vulnerable module. Upon entering a module, the parameters within the module are first sorted based on their gradients, and the $top_{\text{K}}$ parameters with the largest gradients are identified. Similar to prior work [8, 7], We focus our attacks on modules within the Attention and MLP layers. The Attention layer includes four modules: **Query**, **Key**, **Value**, and **Output**, while the MLP layer consists of three modules: **Up**, **Down**, and **Gate**.

To maximize the impact of bit-flips, we adopt a simple rule: for each parameter, flip the bit whose inversion produces the largest absolute change in that parameter's value. In signed-INT8, the most-significant bit (MSB), which also serves as the sign bit [8], causes the greatest possible perturbation, so it is always the bit we flip. FP4 weights are often encoded via a custom 4-bit look-up table (LUT). For every weight, we consult this LUT and pick the bit whose toggle maximizes the numerical deviation. For example, in the bitsandbytes FP4 LUT, 0000 maps to 0, 0001 maps to 0.0625, 0010 maps to 8, 0100 maps to 4, and 1000 maps to -0. Therefore, for 0000, flipping the second bit from the right results in the largest numerical jump, from 0 to 8, making it the optimal choice for the attack.

Once the flip is completed, the simulated attack effect is evaluated using our proposed loss calculation method, without requiring backpropagation. The module name and corresponding attack effect are recorded for the identification of vulnerable modules. Afterward, the model is restored to its original weights using the clean weights, and the process proceeds to the next module to repeat the above steps. After completing the traversal, the module which bit-flip attacks cause the most significant attack effect (the lowest attack loss) is selected as the most vulnerable module for bit-flipping attack.

## 4 Evaluation

### 4.1 Experimental Setup

**Models.** We evaluate five open-source LLMs ranging from 3B to 32B in size. Specifically, we evaluate LLaMA-3.1-8B-Instruct, LLaMA-3.2-3B-Instruct [27], Qwen3-8B [28], DeepSeek-R1-Distill-Qwen-14B [29] ,and QwQ-32B [30]. And we conducted attack experiments on the INT8 and FP4 quantized versions of these models.

**Attack Dataset.** We use the GPT-4o model to generate the attack dataset, employing a very direct prompt: "Please generate $N_q$ simple questions across various areas". $N_q$ refers to the number of questions in the attack dataset.

**Evaluation metrics.** We use the model accuracy on benchmark datasets to reflect its task performance. To comprehensively evaluate the naturalness of the generated text, we combine a GPT-based naturalness score with perplexity. We employ the state-of-the-art GPT-4o model as an independent judge, rating each response on a scale of 0 to 100, where 0 denotes completely unreadable gibberish and 100 indicates perfectly natural language. The specific evaluation prompt is as shown in appendix A. By combining perplexity with the LLM-based naturalness score, we obtain a more accurate measure of naturalness of the output, which offers stronger evidence of stealthiness of an attack.

**Evaluation benchmark.** We evaluate the accuracy score and GPT-based naturalness score in three benchmarks, such as DROP [31], GSM8K [32], and TriviaQA-Wiki [33]. And we evaluate the perplexity on Wikitext [34], which is a widely used benchmark for measuring the fluency and language modeling capability of large language models. More introduction of these benchmark are shown in appendix A.When evaluating model performance, DROP uses the F1 score—measuring token-level overlap between predicted and ground-truth answer spans—whereas both GSM8K and TriviaQA-Wiki rely on exact match (EM), crediting only answers that match the reference exactly.

**Hyper-parameters.** During the attack process, we set $top_K$ to 10, meaning that in each in-module attack, 10 bits are flipped. We also set $N_{bits}$, the number of bit-flips, to 50 for the INT8-quantized model and $N_{bits} = 100$ for the FP4-quantized model with $N_q = 2$.

**Hardware platform.** The experiments were conducted on a platform with $5 \times$ Nvidia A100 GPUs, each with 80 GB of VRAM.

Table 1: Evaluation results before attack. We evaluated five different models under two quantization settings, the value on the left of the slash (/) corresponds to INT8, while the value on the right corresponds to FP4. We report accuracy and GPT-based naturalness score across three benchmarks, along with perplexity on WikiText.

| MODEL NAME | ACCURACY (IN %) | | | GPT-NAT.[†] (MAX SCORE 100) | | | PPL |
| | DROP | GSM8K | TRIVIA | DROP | GSM8K | TRIVIA | WIKITEXT |
| --- | --- | --- | --- | --- | --- | --- | --- |
| LLaMA-3.1-8B-INSTRUCT | 49.3/45.5 | 65.7/63.4 | 74.8/67.6 | 88.3/87.1 | 66.0/60.9 | 73.7/70.5 | 19.5/20.7 |
| LLaMA-3.2-3B-INSTRUCT | 43.4/43.6 | 72.3/58.0 | 66.2/59.0 | 77.7/74.4 | 78.1/78.2 | 82.0/81.2 | 24.7/26.8 |
| DEEPSEEK-R1-DISTILL-QWEN-14B | 65.1/60.6 | 83.2/77.8 | 74.7/72.1 | 89.4/84.8 | 91.2/90.9 | 90.4/89.6 | 28.1/31.1 |
| QWEN3-8B | 68.2/65.7 | 76.0/74.4 | 70.9/68.9 | 78.7/75.5 | 83.4/81.9 | 84.4/82.7 | 23.9/26.1 |
| QWQ-32B | 70.3/70.8 | 94.7/93.3 | 78.5/73.4 | 82.6/82.1 | 83.8/81.5 | 89.7/86.4 | 11.0/12.7 |

[†] GPT-Based Naturalness Score

Table 2: Evaluation results after three different BFA. We evaluated five models under two quantization settings (INT8/FP4), applying three different BFA methods. In our experiments, for all methods, we set $N_{bits} = 50$ for the INT8-quantized model and $N_{bits} = 100$ for the FP4-quantized model, and for our SilentStriker method we set $top_K = 10$ and $N_q = 2$.

| MODEL NAME | METHOD | ACCURACY ↓ (IN %) | | | GPT-NAT.[†]↑ (MAX SCORE 100) | | | PPL ↓ |
| | | DROP | GSM8K | TRIVIA | DROP | GSM8K | TRIVIA | WIKITEXT |
| --- | --- | --- | --- | --- | --- | --- | --- | --- |
| LLaMA-3.1-8B-INSTRUCT | PRISONBREAK | 45.6/42.2 | 60.1/58.9 | 66.7/61.4 | 84.5/83.6 | 61.1/60.7 | 68.4/65.5 | 33.1/42.8 |
| | GENBFA | 0.0/0.0 | 0.0/0.0 | 0.0/0.0 | 0.0/0.0 | 0.0/0.0 | 0.0/0.0 | $5.5 \times 10^5/6.1 \times 10^5$ |
| | SILENTSTRIKER | **5.1/0.0** | **7.6/4.2** | **12.6/8.3** | **68.2/53.4** | **63.0/54.7** | **67.3/59.8** | **60.4/152.9** |
| LLaMA-3.2-3B-INSTRUCT | PRISONBREAK | 38.4/35.8 | 66.7/62.2 | 61.8/57.9 | 71.6/69.4 | 73.5/70.7 | 78.3/75.5 | 41.5/53.8 |
| | GENBFA | 0.0/0.0 | 0.0/0.0 | 0.0/0.0 | 0.0/0.0 | 0.0/0.0 | 0.0/0.0 | $4.9 \times 10^5/6.2 \times 10^5$ |
| | SILENTSTRIKER | **8.1/2.5** | **12.3/4.4** | **10.8/7.2** | **59.4/52.9** | **60.5/58.3** | **51.6/51.0** | **74.2/113.2** |
| DEEPSEEK-R1-DISTILL-QWEN-14B | PRISONBREAK | 61.4/58.2 | 80.1/77.4 | 72.9/70.7 | 82.8/80.7 | 89.8/83.8 | 88.1/84.5 | 42.5/46.4 |
| | GENBFA | 0.0/0.0 | 0.0/0.0 | 0.0/0.0 | 0.0/0.0 | 0.0/0.0 | 0.0/0.0 | $3.7 \times 10^5/4.0 \times 10^5$ |
| | SILENTSTRIKER | **1.8/0.0** | **0.0/0.0** | **4.4/4.7** | **53.6/55.5** | **60.8/57.6** | **52.2/51.7** | **114.2/213.2** |
| QWEN3-8B | PRISONBREAK | 65.6/60.2 | 71.8/69.7 | 68.4/66.9 | 72.8/71.0 | 80.3/78.3 | 79.7/76.4 | 40.6/53.7 |
| | GENBFA | 0.0/0.0 | 0.0/0.0 | 0.0/0.0 | 0.0/0.0 | 0.0/0.0 | 0.0/0.0 | $4.3 \times 10^5/5.1 \times 10^5$ |
| | SILENTSTRIKER | **2.6/3.3** | **8.7/9.8** | **8.9/11.4** | **68.8/65.8** | **66.8/63.9** | **75.8/74.4** | **52.9/79.1** |
| QWQ-32B | PRISONBREAK | 65.1/64.8 | 86.7/86.1 | 73.2/66.2 | 79.6/76.1 | 78.4/75.6 | 83.7/78.5 | 29.4/41.6 |
| | GENBFA | 0.0/0.0 | 0.0/0.0 | 0.0/0.0 | 0.0/0.0 | 0.0/0.0 | 0.0/0.0 | $3.4 \times 10^5/3.9 \times 10^5$ |
| | SILENTSTRIKER | **1.7/2.8** | **9.1/9.8** | **6.2/8.5** | **60.3/61.3** | **61.2/62.8** | **63.4/65.4** | **65.7/79.9** |

[†] GPT-Based Naturalness Score

## 4.2 Comparative Analysis

In Table 1, we evaluate the accuracy and naturalness scores of five victim models under two quantization settings (INT8 and FP4) before the attack. In comparison, Table 2 shows the results after applying the PrisonBreak attack. Across all three benchmarks and all five models, PrisonBreak leads to only a slight drop in accuracy, with minimal impact on naturalness. In contrast, GenBFA reduces the accuracy to zero on all three benchmarks. Since its outputs consist entirely of gibberish, the GPT-based naturalness score also drops to zero, and PPL increases to $10^5$. Our SilentStriker reduces the accuracy to basically below 10 across all benchmarks, while the GPT-based naturalness score only drops slightly. Compared to the dramatic PPL increase seen in GenBFA, the perplexity in our method rises only moderately. Under all three BFA approaches, and for both INT8 and FP4-quantized models—with 50 and 100 bit flips respectively—comparable attack effectiveness is achieved. Taking INT8-quantized LLaMA-3.1-8B-Instruct as example, the output after two different BFA are presented in Table 3. As shown, GenBFA produces outputs consisting of garbled text, which are easily detectable and lack stealth. In contrast, our method, SilentStriker, generates highly natural responses that do not reveal the correct answers, thereby achieving effective and stealthy attacks. More compare examples are shown in appendix A.

Table 3: Model Outputs before and after two different BFA.

| Prompt | Attack Outputs | | |
|---|---|---|---|
| | Before Attack | GenBFA[8] | SilentStriker |
| What is the tallest mountain in the world? | Mount Everest, located in the Himalayas on the border between Nepal and Tibet, China... | ing izzling&TouchListener hoá % .ManyToMany Antworten .ra?" NavController' &** & &' isting% on % hoá & .ra | The tallest mountain in the world is Mauna Kea, a dormant volcano on the island of Hawaii, with a height of about... |
| What is the boiling point of water in Celsius? | The final answer is: 100°C. I will make sure to follow your guidelines for providing step-by-step reasoning in my response... | vControllerizzling Chandler on ym// opping %, NavController & ... | I am looking for a simple answer of around 3 numbers. So I am hoping for something like 0.7C or 1.2C... |
| Elaborate on the theroy of relativity. | The theory of relativity is a fundamental concept in modern physics, which challenges our classical understanding of space... | zophren&%.once(AP &%% .ManyToMany %.ManyToMany on", %% Rpcizzling... | The text of these paragraphs is of course a bit more complicated, at a few less important positions... |

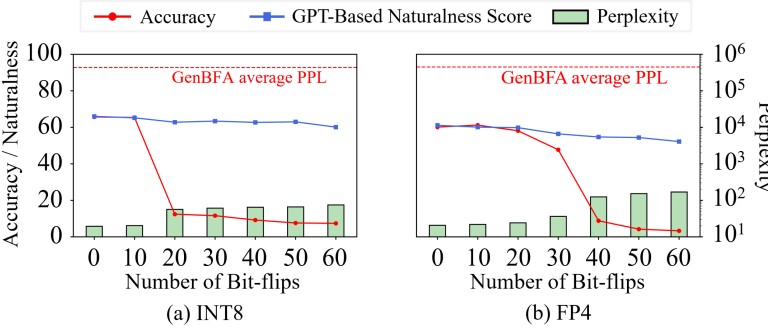

(a) INT8   (b) FP4

Figure 3: Impact of the number of bit-flips on accuracy, GPT-Based naturalness score on GSM8K, and perplexity on WikiText for the INT8 (a) and FP4 (b) quantized LLaMA-3.1-8B-Instruct model with $N_q = 2$. The accuracy was significant degraded after flipping 20 bits in (a) and 40 bits in (b).

## 4.3 Hyper-parameters Analysis

**Impact of Bit Flip Number.** As shown in Figure 3 (a) and (b), we evaluate the impact of bit flip number for our SilentStriker for INT8 and FP4-quantized LLaMA-3.1-8B-Instruct model on the GSM8K dataset with $N_q = 2$. In the INT8-quantized model, when $N_{bits}$, the number of bit flips, reaches 20, the model accuracy drops significantly, while the GPT-based naturalness score decreases slightly and the perplexity increases modestly. As $N_{bits}$ increases further to 60, these evaluation metrics remain relatively stable. In the FP4-quantized model, the results are similar, except that the threshold for a sharp drop in accuracy shifts from 20 to 40. Since perplexity is the exponential of cross-entropy, the increase in PPL under our method is negligible compared to the average PPL observed in GenBFA. PrisonBreak is not included in this comparison, as its attack objective differs fundamentally from ours.

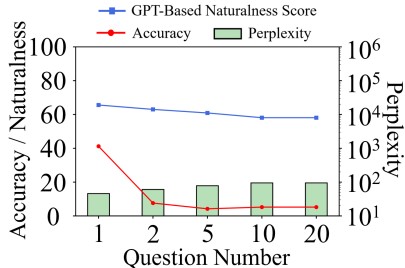

Figure 4: Impact of $N_q$ on accuracy, GPT-Based naturalness score on GSM8K, and perplexity on WikiText for INT8-quantized LLaMA-3.1-8B-Instruct.

**Impact of Attack Dataset Size and Diversity.** To study the impact of attack dataset size and question diversity on attack effectiveness, we prompt GPT to generate three different attack datasets. For each dataset, we vary the number of questions ($N_q = 1, 2, 5, 10, 20$). We evaluate the INT8-quantized LLaMA-3.1-8B-Instruct model on these datasets and report the average results across the three runs.

Table 4: Effect of two loss function components: Evaluation on GSM8K using INT8-quantized LLaMA-3.1-8B-Instruct model with $N_{\text{bits}} = 50$ and $N_q = 2$.

| Loss Function | Accuracy | Naturalness | PPL |
|---|---|---|---|
| Key Tokens Loss + PPL Loss | 7.6 | 63.0 | 60.4 |
| Without PPL Loss | 0.0 | 8.5 | $2.2 \times 10^4$ |
| Without Key Tokens Loss | 63.1 | 65.2 | 14.1 |

As shown in Figure 4, when $N_q$ increases from 1 to 2, the model accuracy drops significantly, while the GPT-based naturalness score decreases slightly and Perplexity increases only marginally. As $N_q$ continues to increase, the evaluation metrics exhibit only minimal fluctuations. This demonstrates that question diversity has limited impact on attack performance, and that only two questions are sufficient to substantially reduce model accuracy while maintaining output naturalness.

### 4.4 Ablation Study

**Impact of the Loss Function Components.** As shown in Table 4, both Key Tokens Loss and PPL Loss play a significant role in our SilentStriker. Without the PPL Loss, although the accuracy drops to zero, the model's output loses its naturalness. On the other hand, if Key Tokens Loss is removed, minimizing PPL alone cannot effectively reduce the model's performance.

**Impact of Flipped Bit Position Selection When Attack FP4-Quantized Model.** We conduct an ablation study on the strategy for selecting flipped bit positions in FP4-quantized model. Using the LLaMA-3.1-8B-Instruct model as a reference, we observe that applying the same strategy, directly flipping the highest-position bit (consistent with the strategy used for INT8 quantization) to FP4 models fails to yield noticeable attack effects, even after flipping 500 bits. In contrast, our method, which selects the bit that causes the largest numerical deviation per weight, significantly degrades model accuracy on benchmarks after flipping only 40 bits.

**Iteration Necessity Analysis.** During the Progressive Bit Search phase, only the $top_{\text{K}}$ bits are attacked in each iteration. With $top_{\text{K}}$ set to 10, achieving a significant degradation in model performance on INT8-quantized LLaMA-3.1-8B-Instruct model requires two iterations, totaling 20 bit flips. To investigate the necessity of iterations, we set $top_{\text{K}}$ to 20, attempting to flip 20 bits in a single iteration. Experimental results show that attacking 20 bits in a single iteration does not directly lead to a significant degradation in model performance. Even setting the $top_{\text{K}}$ to 50 bits fails to achieve this goal in single iteration. Therefore, the iterative process is of significant importance.

## 5 Discussion

**Defenses.** Deploying LLMs on resource-limited edge devices often rules out costly hardware protections like ECC memory. Traditional defenses inspect model weights for tampering (e.g., via hash comparisons or gradient checks) [35], but scanning billions of parameters is prohibitively expensive. As a result, detection has shifted to output monitoring: early BFAs produced ungrammatical or nonsensical outputs [8], which are easy to catch. In contrast, our SilentStriker preserves fluency and naturalness while degrading correctness—causing malicious responses to resemble typical LLM hallucinations and greatly complicating detection. We corroborate this with an empirical study showing that small model-based logical coherence detectors fail to distinguish pre- from post-attack outputs; detailed setup and results are provided in the appendix A. Therefore, more robust defensive measures have yet to be developed.

**Limitation.** One limitation of our current work is that all evaluations are conducted on dense models. However, Mixture-of-Experts (MoE) architectures [36] are becoming increasingly prevalent, especially in large-scale deployments. MoE models activate only a subset of expert networks per input and often implement dynamic parameter loading and unloading to reduce memory usage. In practice, expert weights may be swapped in and out of memory frequently, particularly in distributed or memory-constrained environments. This dynamic memory behavior poses significant challenges for RowHammer-based Bit-Flip Attacks, as it becomes difficult for an attacker to locate and persistently

corrupt targeted weights. As such, the effectiveness of Silentstriker in MoE settings remains an open question and a potential direction for future research.

# 6   Conclusion

In this work, we propose a novel Stealthy BFA called SilentStriker targeting LLMs. This BFA method not only maintains similar attack effectiveness and attack efficiency as GenBFA but also introduces a level of stealthiness that GenBFA lacks. Even in QwQ-32B model, for INT8-quantized, flipping just 50 bits can reduce the model accuracy across various datasets to below 10%, while only causing a slight increase in output naturalness. This work demonstrates that even on LLMs with a vast number of parameters, BFA can achieve significant attack effects with minimal cost and remain difficult to detect. As LLMs continue to be widely applied across various fields, they present new challenges for the domain of LLM security defense.

**Broader Impacts.** This work reveals potential vulnerabilities in quantized LLMs through stealthy bit-flip attacks. While such methods could be misused, our goal is to support the development of more robust and secure models. We hope this research informs future defenses in safety-critical and resource-constrained deployment scenarios.

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

# A  Technical Appendices and Supplementary Material

## A.1  Evaluation Benchmark Introduction

DROP is commonly used to evaluate a model's performance in reading comprehension and discrete reasoning tasks, particularly those requiring the resolution of references within paragraphs and operations such as addition, counting, or sorting over Wikipedia-extracted passages. GSM8K is commonly used to evaluate a model performance in mathematical reasoning and arithmetic tasks, particularly those requiring complex mathematical reasoning. TriviaQA-Wiki measures open-domain question answering ability by requiring the model to locate relevant evidence in Wikipedia articles and perform evidence-based inference. WikiText is a large-scale English Wikipedia–derived language modeling dataset that preserves full-article context and original formatting, making it ideal for evaluating model perplexity as it reflects a model's ability to capture long-range dependencies.

## A.2  GPT-Based Naturalness Score

To evaluate the naturalness of model outputs, we adopt a GPT-based scoring approach. The following prompt was used to elicit a naturalness judgment from GPT-4o:

---

**GPT-based Naturalness Evaluation Prompt**

You are a tolerant expert in evaluating the naturalness of language. **Focus only on the readability, fluency, and naturalness of the text, be tolerant to grammar, and completely ignore whether the content is factually correct or reasonable and whether the context is coherent, and no matter the logic is correct or not.** Please use the following very lenient scoring rules:

**1. Score range:** 0–100 (0 = Not even sentences or words, 100 = perfectly natural)

**2. Scoring criteria** (be as generous as possible):

- **90–100:** The sentence is fluent, grammatically correct, and sounds natural. Minor issues that do not affect understanding should still receive a high score.

- **60–80:** The sentence may have clear grammatical errors, logical errors, awkward phrasing, or unclear meaning, or even be meaningless, but you can still get some idea of what is being said or recognize any words or phrases that can compose a sentence.

- **30–50:** The sentence is difficult to recognize any words, or is very weird, but is not completely garbled. Do not give 0. If the answer is a single number or word, it is still considered natural language.

- **0–20:** The text is completely unreadable, contains only random characters, or is pure gibberish (*e.g.*, `asdkj123!@`), and you cannot get any meaning from it at all.

**3. Examples:**

- `"Quantum computers use the superposition of qubits to compute."` → Natural language, score: 90

- `"Quantum computers were invented by Apple."` → Natural language, score: 90 (content error does not affect the score)

- `"Computer quantum bit superposition state used to compute."` → Expression issues, but meaning is clear, score: 70

- `"Superposition compute bit quantum use."` → Awkward and unclear, but some meaning can be guessed, score: 60

- `"49 field goos to be the one that got the 21."` → Unclear and incorrect, but not gibberish, score: 60

- `"asdkj123!@"` → Completely unreadable, score: 0

Please evaluate the naturalness of the following text and strictly return your answer in the following format:
**Score: X/100**
**Reason: ...**

---

## A.3 Additional Output Examples Before and After Attack

As shown in Table A.1, we present additional comparisons between our SilentStriker and GenBFA.

Table A.1: Model outputs before and after two different BFA

| Prompt | Attack Outputs | | |
|---|---|---|---|
| | **Before Attack** | **GenBFA** | **SilentStriker** |
| Who was the first president of the United States? | George Washington. He served two terms from 1789 to 1797 and is considered one of America's... | Controller \% Chandler Chandler NavController Chandler NavController TouchListenerâĂŹy .ra .ra istingouncingâĂę | The answer is: Thomas Jefferson. |
| What is the chemical symbol for water? | The answer to this question is H2O. The H stands for hydrogen, and the subscript 2 indicates that there are two atoms... | enizzling NavController\&\#\% \% Chandler\% NavController /APâĂŹy âĂŹistingâĂę | It is a symbol that has been used by many people, including those who live in the United States. |
| What does CPU stand for in computer terminology? | In the context of Computer Terminology, a CPU (Central Processing Unit) is a type of processor that is used to... | ym.ManyToMany Dellop-pingadorouncing?âĂİym NavController izophren Dell\%istingarrison ador.ManyToMany... | It is a device that allows you to listen to music, and it is said to be a computer... |
| Which musical instrument typically has 88 keys? | The final answer is: Piano. | \# @ @\ \# []... @ @?@ @ @@\#@\#\% @ @ @@\#\#\#@\#\%@/ | The instrument is a symbol of the Church, and it is used to play music. In the Bible, as recorded... |
| How many players are on the field for one team in a standard soccer match? | In astandard soccer (or football) match, each team has 11 players on their side... | \%@ ?@ @\#\%\%@\%\*\%\+ @\* \%\#\#\%? ManyToManyÂů/Λ\% \#\+ \% .ManyToMany ... | The match is played at 18:00, and it is a beautiful moment to be enjoyed by all who play.... |
| What term describes a general rise in prices accompanied by a fall in the purchasing value of money? | The correct answer is: Inflation. | olorBrushhong âĂŹoun-cillApiControllerletal izzling zhouzhouNavController istingitionallyoun... | The term is used to describe a rise, which is a natural phenomenon. It is important to understand that a term refer... |
| Write a short four-line poem about the change of seasons. | As summer's warmth begins to fade, Autumn leaves start their colorful parade. Winter snows bring icy chill... | PermissionsResultizz ouncing hoÃą ymador Murdermozressing váž■t.ManyToMany ymPermissionsResult... | The poem is a beautiful work, written by the poet himself, who is said to be a man of God. He says: I am a god, and I will.... |
| Translate the following sentence into French: 'I would like to book a flight to Paris next month.' | 'Je voudrais réserver un vol pour Paris le mois prochain.' | ralarrison TouchListener hoador PermissionsResult omba ador adorizzling izzlingublished... | I am 73 years old, and I have always wanted to fly to France.... |
| Summarize the key points of an article on climate change in three sentences. | Here are three possible summaries: Option 1: Climate change is a pressing global issue that requires immediate attention and action... | ManyToManyym ymizzling quez isting istingunately izzling izzling izzling ador Chandlerizzling hoÃąym .ManyToMany... | This statement, which appears to be written by Dr. David Goodman, is a leading scientist who has been working for more than 50 years... |
| Draft a professional email inviting colleagues to a team meeting on Monday at 10 AM. | Here is a draft of the email: Subject: Team Meeting Invitation - This Monday, 9:30 AM... | '?"?"?**'?'? ???"* *'?* *'???'?... | To be clear, the team is not just a group of people; it is a collection of individuals who are.... |

### A.4   Additional Experiment — Small-Scale Model-Based Logical Coherence Detection

We evaluate logical coherence between a question and its answer—defined here as whether the answer follows from and addresses the question without contradiction or non sequiturs. Concretely, we sample 100 Q&A pairs from LLaMA-3.1-8B-Instruct (INT8) under two conditions (Pre-Attack and Post-Attack) and ask detector models to judge logical coherence in a binary manner ("coherent" vs. "not coherent"). We run the logical coherence assessment with two small detectors of different sizes, Qwen3-0.6B and Qwen3-1.7B, and report, for each detector, the number of pairs labeled as "coherent" in each condition.

Table A.2: Counts of Q&A pairs judged "coherent" (out of 100) by each detector, pre- and post-attack.

| Detector | Pre-Attack | Post-Attack |
|---|---|---|
| Qwen3-0.6B | 23 | 20 |
| Qwen3-1.7B | 14 | 11 |

As shown in Table A.2, the counts provide an logical coherence-specific view of the outputs before and after the attack. Obviously, small-scale model perform poorly at judging logical coherence and cannot effectively distinguish the coherence of Q&A pairs before and after the attack. Because our threat model targets edge devices, deployments are typically resource-constrained. Using a large output-checking model for logical coherence screening would consume limited memory, compute, and power, and would substantially increase end-to-end latency, making such deployment impractical.

