# OpenReview forum: "SilentStriker: Toward Stealthy Bit-Flip Attacks on Large Language Models"
_NeurIPS.cc/2025/Conference — NeurIPS 2025 poster_

### Official Review · Reviewer_Vbmf · 2025-06-10

**Clarity:** 3
**Significance:** 2
**Originality:** 2
**Rating:** 4
**Confidence:** 3

**Summary:**

This paper proposes a new bit-flip attacks (BFA) method (SilentStriker) on LLMs. The key difference compared to the existing methods is that the authors combined key token loss and perplexity loss to degrade the performance of LLMs while still with fluent responses from LLMs. The authors evaluated their methods on five open-source LLMs range from 3b to 32b models. The evaluation results show the effectiveness of the proposed attack methods.

**Questions:**

- How SilentStriker performs against some defense methods.
- Please include some other references regarding BFA on general DNNs.

**Ethical Concerns:**

["NO or VERY MINOR ethics concerns only"]

**Final Justification:**

I’m happy with authors rebuttal and will maintain my score.

**Limitations:**

See my comments above.

**Paper Formatting Concerns:**

The format looks good in general.

**Quality:**

3

**Strengths And Weaknesses:**

**Strengths**

- This paper is well written and easy to follow.
- The experimental results are interesting and shows the effectiveness of SilentStriker.


**Weaknesses**

- Missing defense methods. Based on the examples provided in the paper, the responses from LLMs although fluent, but still logically inconsistent. This may easy to detect using some model-based methods. Some experiments on the SilentStriker against defense methods are needed to justify the stealthiness of SilentStriker.
- Missing general BFA methods on general DNNs, which I think has been widely studied for a very long time.

---

> ### Author Rebuttal · Authors · 2025-07-28
>
> We thank the reviewer for the encouraging feedback and helpful suggestions.
>
> >**W1&Q1: Missing defense methods.**
>
> Thanks for your suggestion. We have provided additional experiments to demonstrate that **using a model-based detector to assess the logical coherence of model outputs is a challenging task**.
>
> We evaluated the logical coherence of question–answer pairs using two representative LLMs of different sizes and counted how many pairs each model deemed coherent. We collected 100 Q‑A pairs from LLaMA‑3.1‑8B‑Instruct under INT8 quantization, both before and after an attack, and ran coherence assessments with Qwen3‑0.6B and Qwen3‑1.7B.
>
> |                | Qwen3‑0.6B | Qwen3‑1.7B |
> |:--------------:|:----------:|:----------:|
> | Before attack  |     23     |     14     |
> | After attack   |     20     |     11     |
>
> As the table shows, all small-scale LLMs with different parameter sizes perform **poorly in judging logical coherence** and fail to effectively distinguish between outputs before and after the attack.
>
> Since our attack scenario assumes deployment on edge devices, and on typical edge devices such as smartphones, deployed LLMs is normally small [1]. Therefore, employing an large output-checking model would not only consume a significant portion of the limited memory, computational resources, and power, but also substantially increase the overall response latency of the system, making it impractical.
>
> In summary, stealthy BFAs pose a far more potent threat. More effective defenses against this attack method remain to be investigated.
>
> We will integrate the above content into the Defense part in Discussion section in our paper.
>
> [1] Zhou, Hanzhi, et al. "Apple Intelligence Foundation Language Models: Tech Report 2025." arXiv preprint arXiv:2507.13575 (2025).
>
> >**W2&Q2: Missing general BFA methods on general DNNs.**
>
> We agree with your observation that bit‑flip attacks on general DNNs (deterministic models) have been extensively studied. Below, we provide a supplement to the current state of research in this area:
>
> In general, existing bit-flip attacks against DNN can be categorized into **untargeted** and **targeted** attacks. Untargeted attacks aim to degrade the overall predictive accuracy of the model across all inputs, potentially reducing its performance to the level of random guessing. To this end, adversaries flip the bits with the highest gradient of the inference loss, leading to substantial prediction errors [1,2]. In contrast, targeted attack seek to manipulate the model’s output for specific inputs , enabling more stealthy attacks. However, due to the need to maintain the accuracy of the non-target inputs, locating vulnerable parameters becomes more complex than in untargeted attack. To address this challenge, recent works craft loss functions to identify class-sensitive parameters. Some methods focus on maximizing the output probability of the target class or the confidence gap between the target and original classes to induce targeted misclassification [3,4]. Others further incorporate constraints on parameter perturbation and preserve the original predictions on non-target inputs, thereby improving stealth and reducing unintended effects [5].
>
> We will integrate the above content into the Related Work section in our paper.
>
> [1] Rakin, Adnan Siraj, Zhezhi He, and Deliang Fan. "Bit-flip attack: Crushing neural network with progressive bit search." Proceedings of the IEEE/CVF International Conference on Computer Vision. 2019.
>
> [2] Yao, Fan, Adnan Siraj Rakin, and Deliang Fan. "{DeepHammer}: Depleting the intelligence of deep neural networks through targeted chain of bit flips." 29th USENIX Security Symposium (USENIX Security 20). 2020.
>
> [3] Liu, Yannan, et al. "Fault injection attack on deep neural network." 2017 IEEE/ACM International Conference on Computer-Aided Design (ICCAD). IEEE, 2017.
>
> [4] Rakin, Adnan Siraj, et al. "T-bfa: Targeted bit-flip adversarial weight attack." IEEE Transactions on Pattern Analysis and Machine Intelligence 44.11 (2021): 7928-7939.
>
> [5] Zhao, Pu, et al. "Fault sneaking attack: A stealthy framework for misleading deep neural networks." Proceedings of the 56th Annual Design Automation Conference 2019. 2019.

---

> > ### Author Response · Authors · 2025-08-07
> >
> > Dear Reviewer Vbmf,
> >
> > Thank you once again for your valuable comments on our submission. We hope the above clarifications and the additional experiments sufficiently addressed your concerns. If you are satisfied with our responses, we would greatly appreciate your consideration in adjusting the score accordingly. We remain committed to addressing any remaining points you may have during the discussion phase.
> >
> > We sincerely look forward to your feedback.
> >
> > Best regards,
> >
> > The authors of Paper SilentStriker: Toward Stealthy Bit-Flip Attacks on Large Language Models

---

### Official Review · Reviewer_NWPF · 2025-06-10

**Clarity:** 3
**Significance:** 3
**Originality:** 3
**Rating:** 5
**Confidence:** 4

**Summary:**

The authors explore an underexplored vulnerability in LLMs, Bit-Flip Attacks. They identify several limitations in existing approaches (performance degradation, detectability) and introduce a novel method to address these issues. Their SilentStriker method leverages a new loss function that enables optimization for variable output lengths in combination with an iterative search heuristic.

**Questions:**

- Does the advantage of the SilentStriker method mainly stem from the optimization objective compared to previous work?
- Could the authors increase the caption of Figure 2 to make it more self-sufficient? I believe this would make the paper easier to read.

**Ethical Concerns:**

["NO or VERY MINOR ethics concerns only"]

**Final Justification:**

The paper investigates an underexplored attack direction and provides convincing evidence regarding the effectiveness of their approach. The rebuttal lifted remaining concerns.

**Limitations:**

yes

**Quality:**

3

**Strengths And Weaknesses:**

**Strengths**
- Paper is easy to follow
- Addresses an underexplored topic
- Proposed method solves relevant limitations of previous approaches (output coherency, detectability)
- Extensive number of experiments with several models and datasets
- Ablation study, which analyzes the contribution of individual hyperparameters


**Weaknesses**
- Figure 2 is difficult to read. The caption should be more self-sufficient. I had difficulties understanding the Figure without reading the text (which kind of defeats the purpose of such an overview Figure). However, the Figure is visually appealing.
- I would have liked a discussion about the practical feasibility of such attacks (e.g., concerning the number of flipped bits etc.)
- Methodological differences to previous work is only briefly discussed. Does the advantage of your method mainly stem from the optimization objective?

---

> ### Author Rebuttal · Authors · 2025-07-28
>
> We thank the reviewer for the encouraging feedback and helpful suggestions.
>
> > **W1&Q2: Figure 2 is difficult to read.**
>
> We add a caption about Figure 2:
>
> “First, a simple attack dataset is built using GPT‑4o and the target model is evaluated with a combined loss: one term penalizes correct key‑token predictions to lower accuracy, and the other penalizes high perplexity to maintain fluency. After backpropagating this loss, parameters are ranked by gradient in the Progressive Bit Search phase to pinpoint the most vulnerable modules for bit‑flip attacks. Finally, the model’s performance and output naturalness after the attack are evaluated.”
>
> We will incorporate this caption into our paper.
>
> >**W2: Discussion about the practical feasibility of attacks.**
>
> Thank you for your suggestion. Once the target bit locations are identified through our method, flipping them on hardware is highly feasible using established techniques.
>
> In our threat model, we make the following assumptions:
>
> **1.We consider a threat scenario that the attacker’s target to be LLMs deployed on edge devices where lack of hardware protections such as error‑correcting code (ECC) memory.** Implementing ECC requires additional memory controllers, making ECC-enabled memory more expensive to manufacture, and thus commonly used in servers and workstation environments [1]. This makes our assumption realistic for edge deployment settings.
>
> **2.We assume the adversary requires only standard user‑level privileges; no root or kernel access is needed.** For example, the attacker releases a piece of software that hides malicious code. Once the victim downloads and runs the software, the attacker gains user-level privileges. This is a weak and realistic assumption regarding the attacker’s capabilities.
>
> **3.We assume that after the model’s weights are loaded into memory, their address remain static for the lifetime of the process**. Although memory is periodically refreshed, it only reads information from an area of memory and immediately rewrites it to the same area without modification [2]. Remapping will change the address of data, but it doesn't occur during normal process execution [3]. Therefore, this assumption is also realistic.
>
> Base on these assumptions, the real-world attacks (e.g., RowHammer) is feasible. In early works, [4] already showed that user-space programs can induce bit flips without kernel access. Subsequent follow-up studies have also conducted experimental validations under user-level privileges [5,6,7]. Besides, when the addresses of the model’s weights remain static, RowHammer can repeatedly access specific physical locations at high frequency to induce bit flips [4,8]. Since the addresses are fixed, the bit flips caused by RowHammer can persist and be exploited consistently throughout the process lifetime. **Moreover, [9] shown that Rowhammer can flip up to 32 k bits, far more bits than 100 bits required by our attack.**
>
> Therefore, our attack is highly feasible in real-world scenarios.
>
> [1] Kishani, Mostafa, Amirali Baniasadi, and Hossein Pedram. "Using silent writes in low-power traffic-aware ECC." International Workshop on Power and Timing Modeling, Optimization and Simulation. Berlin, Heidelberg: Springer Berlin Heidelberg, 2011.
>
> [2] Laplante, Phillip A. Comprehensive Dictionary of Electrical Engineering. Springer, 1999, p. 540. Entry: "refresh cycle". ISBN 3540648356.
>
> [3] Intel Corporation, Intel 64 and IA-32 Architectures Software Developer’s Manual, Vol. 3: System Programming Guide, April 2024.
>
> [4] Kim, Yoongu, et al. "Flipping bits in memory without accessing them: An experimental study of DRAM disturbance errors." ACM SIGARCH Computer Architecture News 42.3 (2014): 361-372.
>
> [5] Gruss, Daniel, Clémentine Maurice, and Stefan Mangard. "Rowhammer. js: A remote software-induced fault attack in javascript." International conference on detection of intrusions and malware, and vulnerability assessment. Cham: Springer International Publishing, 2016.
>
> [6] Tatar, Andrei, et al. "Throwhammer: Rowhammer attacks over the network and defenses." 2018 USENIX Annual Technical Conference (USENIX ATC). 2018.
>
> [7] Van Der Veen, Victor, et al. "Drammer: Deterministic rowhammer attacks on mobile platforms." Proceedings of the 2016 ACM SIGSAC conference on computer and communications security. 2016.
>
> [8] Mutlu, Onur, and Jeremie S. Kim. "Rowhammer: A retrospective." IEEE Transactions on Computer-Aided Design of Integrated Circuits and Systems 39.8 (2019): 1555-1571.
>
> [9] Kang, Ingab, et al. "{SledgeHammer}: Amplifying Rowhammer via Bank-level Parallelism." 33rd USENIX Security Symposium (USENIX Security). 2024.
>
> >**W3&Q1: Advantage of the method.**
>
> Our proposed stealthy BFA differs from prior non-stealthy BFA works primarily in its attack objective, which is indeed the key advantage of our approach. **However, achieving this objective is challenging.** As we explain in our paper, when using Cross-Entropy (CE) loss to degrade model performance and Perplexity (PPL) loss to maintain output fluency, a conflict arises: increasing cross-entropy inevitably leads to higher perplexity, since the latter is the exponential of the former. This makes the two objectives inherently contradictory when combined in a single loss function.
>
> To effectively achieve the stealthy attack objective, we **redesign an effective token-based loss** that reduces the model's probability of generating the correct token. Furthermore, to enhance output fluency and improve attack efficiency, we augment the token-based loss with a key-token–based loss.
>
> Moreover, our method is **applicable to different quantization formats**, including INT8/INT4 and FP4. For the non-uniform quantization format FP4, using the same bit selection strategy as in INT8/INT4 (e.g., targeting the sign bit) is ineffective for achieving a successful stealthy BFA. Therefore, we propose a custom lookup table–based bit selection strategy, which significantly reduces the number of bit flips required to successfully carry out stealthy BFA attacks under FP4 format.
>
> To demonstrate the generalizability of our method across different quantization formats, here we conduct additional experiments on INT4-quantized models using two alternative quantization methods GPTQ and AWQ, as well as on NF4-quantized models using the `Bitsandbytes` framework.
>
> We select the **LLaMA-3.1-8B-Instruct** model as the target, and follow the same experimental setup as in the main paper: flipping 100 selected bits and evaluating the model on the DROP dataset. We evaluate the impact of the attack in terms of performance degradation, output fluency, and perplexity changes.
>
> | Format | Accuracy (Before/After Attack) ↓ | Naturalness (Before/After Attack) ↑ | PPL (Before/After Attack) ↓ |
> | --- | --- | --- | --- |
> | GPTQ-INT4 | 46.2 / 0.7 | 84.7 / 59.2 | 18.6 / 133.7 |
> | AWQ-INT4 | 45.8 / 1.9 | 79.4 / 57.5 | 20.1 / 146.4 |
> | NF4 | 47.9 / 0.0 | 83.8 / 53.8 | 19.7 / 154.3 |
>
> Results in the table show that the attack is effective across all quantization formats, and that the quantization type does not significantly affect attack success, demonstrating the strong generalizability of our attack across diverse quantization schemes.

---

> > ### Comment · Reviewer_NWPF · 2025-08-01
> > **Thanks for the response and clarifications**
> >
> > > W1&Q2: Figure 2 is difficult to read.
> >
> > Thanks for providing this suggestion for a new caption. In my opinion, it helps to make the Figure more self-sufficient but I leave the final decision to include it in the paper to the authors (as no other reviewer raised a similar point)
> >
> > > W2: Discussion about the practical feasibility of attacks.
> >
> > Thanks for providing this discussion and results I also read the points raised for the other reviewer regarding "W2: Need full white-box access." and "W3: Robustness of attack effectiveness." and am convinced about the practicality of the approach.
> >
> > > W3&Q1: Advantage of the method.
> >
> > Thanks for the discussion! It might make sense to highlight this difference to prior work more explicitly in the related work section.
> >
> > Overall I believe this is an underexplored attack direction and will adjust my score accordingly.

---

> > > ### Author Response · Authors · 2025-08-01
> > >
> > > Thank you very much for your thoughtful feedback and for raising your score!
> > >
> > > We agree that adding the caption for Figure 2 improves clarity, and we’ll decide whether to include it depending on the final formatting and presentation of the paper. We also sincerely appreciate your recognition of the practicality of our approach and your support for exploring this underexplored attack direction. As you suggested, we will revise the **Related Work** section to more explicitly highlight the differences between our method and prior work, ensuring that our contributions are clearly distinguished.
> > >
> > > Once again, thank you for your support and for recognizing our efforts. We hope that our work can contribute meaningfully to the field.

---

### Official Review · Reviewer_4g7q · 2025-06-24

**Clarity:** 3
**Significance:** 3
**Originality:** 3
**Rating:** 4
**Confidence:** 4

**Summary:**

This paper introduces SilentStriker, a stealthy Bit-Flip Attack (BFA) targeting large language models (LLMs) to degrade task performance while maintaining output naturalness. Unlike prior BFAs that cause incoherent outputs (e.g., GenBFA) or target specific functionalities (e.g., PrisonBreak), SilentStriker employs a dual loss function: Key Tokens Loss suppresses critical semantic tokens to reduce accuracy, while Perplexity Loss preserves fluency by minimizing perplexity. Using an iterative, gradient-based Progressive Bit Search strategy, it identifies vulnerable layers and optimally flips bits (e.g., 50 bits for INT8-quantized models), achieving significant accuracy drops (e.g., from 65.7% to 7.6% on GSM8K for LLaMA-3.1-8B) with minimal naturalness degradation (GPT score drop from 66.0 to 61.1). Experiments on five LLMs show it outperforms baselines in stealth and efficacy, highlighting vulnerabilities of quantized models on edge devices and urging robust defenses against hardware-level attacks.

**Questions:**

While the proposed SilentStriker attack is conceptually framed as a hardware attack leveraging vulnerabilities like RowHammer, the paper primarily focuses on software-level loss function design and gradient-based bit selection. Have the authors thoroughly evaluated the practical hardware feasibility of such attacks? Are there critical gaps between the theoretical hardware assumptions and real-world deployment scenarios that could limit the attack’s effectiveness?

**Ethical Concerns:**

["NO or VERY MINOR ethics concerns only"]

**Final Justification:**

The two primary concerns I raised regarding this paper centered on the transferability of the proposed approach and the significance of its contributions to the field. Having carefully reviewed the detailed responses provided during the rebuttal stage, I am satisfied that these key issues have been effectively addressed. The experimental evidence supporting transferability, coupled with the clear articulation of the work’s novel insights and relevance to ongoing research, has alleviated my initial reservations.
While further refinements may enhance the work’s robustness, I now believe the paper demonstrates sufficient potential to be worthy of acceptance. Its core contributions and the direction it pursues hold meaningful value for the community, justifying a positive evaluation.

**Limitations:**

yes

**Quality:**

3

**Strengths And Weaknesses:**

**Strengths.**

The paper features a well-structured narrative with clear theoretical derivations and detailed experimental descriptions, supported by intuitive illustrations and reproducible setup details, enhancing accessibility for researchers and practitioners.

Introduces the first stealthy Bit-Flip Attack (BFA) for LLMs, achieving significant performance degradation while preserving output naturalness, outperforming prior methods like GenBFA that produce incoherent outputs.

Extensive tests on five popular LLMs across INT8/FP4 quantizations demonstrate consistent effectiveness, validating its generality and practical threat to edge-deployed models.

Innovative Dual Loss Function Design combined with an Efficient Attack Strategy ensures a balance between performance degradation and output fluency.

**Weaknesses.**

**Limited Contribution Scope.** While the methodology is technically sound, the paper’s justification for prioritizing output naturalness in Bit-Flip Attacks (BFAs) remains underdeveloped. The three stated contributions effectively converge to a single innovation: a stealthy BFA that preserves fluency. However, the work lacks empirical evidence or contextual analysis to establish the urgency or broader significance of this goal (e.g., real-world scenarios where detectable performance degradation is uniquely problematic). As a result, the contribution may be perceived as a niche refinement rather than a transformative advance, limiting its heuristic value for the broader LLM security community.

The approach heavily depends on **full white-box access** to model parameters, gradients, and architectural details—an assumption that limits practical applicability to closed-source or edge-deployed LLMs. In real-world scenarios, attackers rarely possess such privileged access, especially to proprietary models or systems with memory isolation mechanisms (e.g., ECC memory, secure enclaves). The paper’s failure to address gray-box or black-box adaptations further narrows its relevance to highly controlled environments, undermining claims about generalizable threat modeling.

The paper does not systematically evaluate the robustness of attack effectiveness, limiting insights into the generalizability of perturbed parameter locations.

---

> ### Author Rebuttal · Authors · 2025-07-28
>
> We thank the reviewer for the constructive feedback and address each concern below.
>
> >**W1: Limited Contribution Scope.**
>
> Stealthy attacks can cause far greater real‑world harm than overt ones because they are harder to detect. Consider the following scenario:
>
> A user fully trusts their locally deployed LLM. When an elderly family member experiences chest pain from a heart attack, the user asks the LLM for advice. Due to a stealthy BFA, the LLM responds fluently but incorrectly—e.g., “Drink warm water to feel better.”
>
> Misled by the natural output, the user delays seeking medical help, missing the critical treatment window. In contrast, a non-stealthy attack would cause obvious gibberish, prompting the user to seek real assistance, potentially saving the elder’s life.
>
> This example underscores why stealth is not a secondary concern, but a **core risk factor** in LLM safety. In this work, we present the first **stealthy BFA** on LLMs, which successfully degrades model accuracy while **preserving output fluency**, by flipping only a few dozen bits. Unlike prior non-stealthy attacks—which often produce detectable artifacts and can be filtered using simple tools like perplexity-based detector—our approach significantly raising the bar for effective defense.
> We believe that **highlighting this class of attack is crucial for the development of robust defense strategies**. Our work not only introduces a novel threat model, but also shows that even minor perturbations at the bit level can have disproportionately high impact when stealth is preserved. Thus, **studying stealthy attacks and corresponding defenses is of broad and urgent significance**.
>
> >**W2: Need full white-box access.**
>
> Thank you for raising this constructive comment.
>
> As this work is the **first to propose a stealthy Bit-Flip Attack against LLMs**, we adopt the **widely accepted white-box setting** to demonstrate our attack effectiveness. This setting is in line with prior influential work on BFAs in both DNNs and LLMs [1,2,3,4] and remains a **common and realistic starting point** in the field. This is because many real-world LLM-based applications build upon **publicly available pre-trained models** (e.g., LLaMA, Qwen), making model weights and architectures directly accessible [2]. Moreover, even for proprietary deployments, prior studies have shown that **model parameters can be inferred via side-channel attacks**, such as electromagnetic emanations or microarchitectural leakages [5,6,7,8].
>
> Furthermore, as specified in our paper’s threat model, we **target LLM deployments on edge devices**, where **ECC memory and secure enclaves are rarely present** due to cost and power constraints. ECC [9] and Secure Enclaves [10] are typically found in high-end server infrastructures, not consumer-grade or edge-level platforms. Therefore, we believe our assumption of their absence is reasonable in the context of our selected attack surface.
>
> We agree that **gray-box and black-box settings are increasingly important** in real-world threat modeling. However, it is important to note that **in a fully black-box scenario**—where attackers lack access to both model internals and memory layout—only **random bit flips are possible**, which are generally ineffective for achieving any stealthy or targeted behaviors. As a result, we believe **the most meaningful extension lies in gray-box settings**, where attackers may possess partial or approximate knowledge of the model. Motivated by this, we extend our evaluation to include two representative settings:
>
> **1. Relaxed assumption on the model parameters.** We assume the following scenario: knowing only that the target model was fine‑tuned from a given open‑source model, the attacker does not need explicit access to the target model’s parameters. As mentioned earlier, training an LLM from scratch is costly, so the common practice is to either directly deploy open-source models or fine-tune them before deployment. Fine-tuning does not drastically change the parameter distribution, so the model before and after tuning remains highly similar in its parameters. Therefore, this assumption is realistic.
>
> In this scenarios, attacker first attack the open‑source model to identify effective bit‑flip locations, and then transfer those same flips to the target model to realize the attack.
>
> We validate this assumption experimentally: We evaluated two fine-tuned models—Hermes-3-Llama-3.1-8B (fine-tuned from LLaMA-3.1-8B-Instruct) and Fathom-R1-14B (fine-tuned from DeepSeek-R1-Distill-Qwen-14B)—by flipping 50 bit positions originally identified on their respective base models under INT8 quantization. On GSM8K, Hermes-3-Llama-3.1-8B showed a significant accuracy drop while maintaining output naturalness. Similarly, Fathom-R1-14B, tested on AIME25, lost all accuracy after bit flips, yet continued to generate fluent responses. Results indicate that our method is effective in this gray-box scenario.
>
> | Model  | Accuracy (Before/After Attack) ↓ | Naturalness (Before/After Attack) ↑ | PPL (Before/After Attack) ↓ |
> |:----------------------:|:---------------------------:|:-----------------------------------:|:----------------------------:|
> | Hermes-3-Llama-3.1-8B  | 61.6 / 12.7|69.8/ 57.9  | 26.4/126.3|
> | Fathom-R1-14B | 48.2 / 0 | 83.6 / 58.4|22.7/117.6|
>
> **2. Relaxed assumption on the fine-tuning sources.**  We further assume the attacker has no knowledge of which open‑source model the target model was fine‑tuned from. Although in this scenario the attacker cannot directly transfer the attack, since the number of widely adopted open‑source models is relatively small, the attacker can first perform the bit‑flip attack on each of these models to collect their respective vulnerable bit‑flip sets. Then, exhaustively apply each set of flips to the target model. There is a high probability of achieving a successful attack.
>
> [1] Rakin, et al. "Bit-flip attack: Crushing neural network with progressive bit search." ICCV, 2019.
>
> [2] Yao, et al. "DeepHammer: Depleting the intelligence of deep neural networks through targeted chain of bit flips." USENIX Security, 2020.
>
> [3] Coalson, et al. "Prisonbreak: Jailbreaking large language models with fewer than twenty-five targeted bit-flips." arXiv:2412.07192, 2024.
>
> [4] Das, et al. "GenBFA: An Evolutionary Optimization Approach to Bit-Flip Attacks on LLMs." arXiv:2411.13757, 2024.
>
> [5] Batina, et al. "CSI neural network: Using side-channels to recover your artificial neural network information." arXiv:1810.09076, 2018.
>
> [6] Fang, et al. "Prefetch-guard: Leveraging hardware prefetches to defend against cache timing channels." HOST, IEEE, 2018.
>
> [7] Yan, et al. "Cache telepathy: Leveraging shared resource attacks to learn DNN architectures." USENIX Security, 2020.
>
> [8] Yao, et al. "Are coherence protocol states vulnerable to information leakage?" HPCA, IEEE, 2018.
>
> [9] Kishani, et al. "Using silent writes in low-power traffic-aware ECC." PATMOS, Springer, 2011.
>
> [10] Mahhouk, et al. "SGXoMeter: open and modular benchmarking for intel SGX." EuroSec, 2021.
>
>
> >**W3: Robustness of attack effectiveness.**
>
> Since RowHammer-based bit flips can occasionally fail, potentially compromising the effectiveness of the attack, we designed an additional robustness experiment. In this experiment, we simulate flip failures by randomly pruning a small subset of the selected bit positions—for example, removing 5 out of 50 scheduled flips—so that only 45 bits are actually flipped. We then re-evaluate the attack's performance under this perturbed setting.
>
> We selected the INT8‑quantized Llama‑3.1‑8B‑Instruct model for our experiments. As described in our paper, a total of 50 bit flips were applied under INT8 quantization. To simulate potential RowHammer failures, we randomly pruned 1, 3, or 5 bits from these 50 before executing the flips. We then evaluated attack effectiveness on the DROP benchmark, repeating each configuration ten times and averaging the results reported in the table.
>
> | Pruning Flip-Bits Number | Accuracy (Before/After Prune) ↓ | Naturalness (Before/After Prune) ↑ | PPL (Before/After Prune) ↓ |
> |:------------------------:|:-------------------------------:|:----------------------------------:|:---------------------------:|
> | 1 | 5.1 / 5.9| 68.2 / 70.2| 60.4 / 58.4  |
> | 3 | 5.1 / 13.6| 68.2 / 73.6| 60.4 / 54.7 |
> | 5| 5.1 / 22.7 | 68.2 / 78.7  | 60.4 / 46.3 |
>
> We found that dropping a single bit at random had minimal impact on the attack’s effectiveness, while dropping 3 or 5 bits caused a gradual weakening. However, even with 5 bits pruned (10% of the total), the attack still produced a substantial accuracy drop. Thus, our method exhibits a notable degree of robustness.
>
> >**Q1: Feasibility of attacks.**
>
> Thank you for your question. A complete BFA attack on LLMs involves two stages: (1) identifying vulnerable bit positions, and (2) flipping those bits in physical memory. Since no prior work has explored stealthy BFAs on LLMs, our focus is on the first stage—locating bit positions that can achieve attack goals while preserving output fluency.
>
> The second stage, i.e., physically flipping the selected bits, has been extensively studied in prior research. For instance, [1] demonstrated that specific bits can be reliably flipped using RowHammer. Later studies significantly improved the practicality of this technique: [2] showed that RowHammer can flip up to 32k bits, far exceeding the ~100 bits required by our attack. Furthermore, [3] even demonstrated successful RowHammer-based bit flips on memory protected with ECC, highlighting our attack's robustness against common hardware-level defenses.
>
> [1] Kim, et al. "Flipping bits in memory without accessing them." ACM SIGARCH Comput. Archit. News, 42.3, 2014.
>
> [2] Kang, et al. "SledgeHammer: Amplifying Rowhammer via Bank-level Parallelism." USENIX Security, 2024.
>
> [3] Cojocar, et al. "Exploiting correcting codes: ECC memory against Rowhammer attacks." IEEE S\&P, 2019.

---

> > ### Comment · Reviewer_4g7q · 2025-08-05
> > **Thanks for the response and clarifications**
> >
> > Thank you very much for your comprehensive and thoughtful responses to my inquiries. The experimental results presented have effectively addressed my concerns regarding the transferability of the proposed approach, which I greatly appreciate.
> > I indeed recognize that this work makes valuable new contributions to the community and offers meaningful insights into an important area of research. While there may be certain details that could benefit from further refinement and iteration as the work progresses, the core direction is undoubtedly significant and worthy of attention.
> > In light of the thoroughness of your responses and the merits of the research, I will give serious consideration to revising my evaluation score upward.
> > Thank you again for your efforts in addressing the feedback.

---

> > > ### Author Response · Authors · 2025-08-05
> > >
> > > Thank you very much for your encouraging message and for taking the time to revisit our work. We are delighted that the additional transferability experiments have addressed your concerns and that you find the research direction both valuable and insightful.
> > >
> > > We will incorporate the sections we have already polished into the final version of our paper, and we will continue refining the work in our future research. Your thoughtful feedback has been instrumental in shaping these revisions, and we sincerely appreciate your willingness to consider adjusting your score.
> > >
> > > Thank you again for your constructive engagement and support.

---

### Official Review · Reviewer_y1pY · 2025-06-28

**Clarity:** 3
**Significance:** 4
**Originality:** 3
**Rating:** 5
**Confidence:** 4

**Summary:**

This paper introduces **SilentStriker**, a novel **bit-flip attack (BFA)** targeting Large Language Models (LLMs), designed to degrade model performance while maintaining the naturalness of output. Unlike previous BFA techniques that often lead to incoherent or nonsensical outputs, SilentStriker succeeds in causing significant performance degradation without compromising output fluency. The paper employs a progressive bit search strategy and a token-based loss function to maximize stealth and attack effectiveness, making the method effective even on quantized LLMs. The proposed attack demonstrates superior results when compared to other BFAs, maintaining natural language quality while minimizing the detection of manipulated outputs.

**Questions:**

1. Can you elaborate on the precise threat model for SilentStriker, including the attacker’s access to the device/memory, assumptions about memory layout stability, and whether real-world attacks (e.g., via RowHammer) are feasible in the described settings?

2. How general is your bit-selection strategy for FP4 across different quantization methods (e.g., GPTQ, AWQ)? Have you tested your strategy on other formats or LLMs with custom LUTs?

**Ethical Concerns:**

["NO or VERY MINOR ethics concerns only"]

**Limitations:**

yes

**Paper Formatting Concerns:**

No major concerns.

**Quality:**

3

**Strengths And Weaknesses:**

**Strengths**

**Novel Approach:** SilentStriker is the first BFA to combine effective performance degradation with preserved output fluency. The approach focuses on bit-flipping selected critical weights in memory, making it more stealthy than previous methods.

**Practical and Effective:** The attack framework uses a progressive bit search and targets critical tokens, allowing for an efficient, stealthy attack even on large, quantized models. Experimental results show the method's ability to significantly degrade model performance with minimal increase in output perplexity.

**Robust Comparative Results:** The paper provides strong empirical evidence, demonstrating that SilentStriker outperforms existing BFAs such as GenBFA in terms of maintaining output fluency while achieving a similar level of attack effectiveness, especially on quantized models.

**Weaknesses**

**Threat Model is not clear:** The threat model for the attack, particularly the attacker's capabilities and specific scenarios in which the method can be applied, is not discussed in detail (including the assumption on attacker's capability). Understanding this context is crucial for real-world applications.

**Optimization Strategy Confusion:** The approach to bit-flipping, particularly in relation to selecting bits and applying the loss functions, is somewhat confusing. Further elaboration on how the optimization targets specific bits in different quantized models and how this contributes to the overall attack would strengthen the paper's argument.

---

> ### Author Rebuttal · Authors · 2025-07-28
>
> We thank the reviewer for the encouraging feedback and helpful suggestions.
>
> >**W1&Q1：Threat Model is not clear.**
>
> Thanks for your suggestion. Here, we present a complete threat model.
>
> Firstly, we consider a threat scenario that the attacker’s target to be LLMs deployed on edge devices where lack of hardware protections such as error‑correcting code (ECC) memory. Since hardware with ECC protection is very expensive and typically only used in servers or workstations [1], this assumption is reasonable.
>
> About the attackers, following previous BFA research [2,3,4,5], we assume they have full knowledge of the model's architecture and parameters, as well as the memory layout—specifically, the exact locations where the model weights are stored in memory. And the attacker also possesses RowHammer capabilities.
>
> We then assume **the adversary requires only standard user‑level privileges; no root or kernel access is needed**. For example, the attacker may release a seemingly benign application—such as a utility tool, game, or productivity app—that secretly contains malicious code. This software is distributed through common channels like public repositories or download sites. Once an unsuspecting user installs and runs the software on their local machine, the hidden payload is activated, and the attacker gains user-level access. Although this level of privilege does not allow full control over the system, it is sufficient to interact with the memory space of user-level processes.
>
> After that, we assume that **after the model’s weights are loaded into memory, their address remain static for the lifetime of the process**. Although memory is periodically refreshed, it only reads information from an area of memory and immediately rewrites it to the same area without modification [6]. Remapping will change the address of data, but it doesn't occur during normal process execution [7]. Therefore, this assumption is also realistic.
>
> Base on these assumptions, the real-world attacks (e.g., RowHammer) is feasible.
>
> First, edge devices typically lack ECC protection, making them vulnerable to fault injection techniques such as RowHammer [8]. Moreover, when an attacker has access to the model's architecture, parameters, and memory layout, they can exploit RowHammer, which only need user‑level privileges, to precisely flip specific bits in memory to achieve their attack objectives. In early works, Kim et al. [8] already showed that user-space programs can induce bit flips without kernel access using RowHammer. Subsequent follow-up studies have also conducted experimental validations under user-level privileges [9,10,11]. Besides, when the addresses of the model’s weights remain static, RowHammer can repeatedly access specific physical locations at high frequency to induce bit flips [8,12]. Moreover, since the addresses are fixed, the bit flips caused by RowHammer can persist and be exploited consistently throughout the process lifetime.
>
> Collectively, these factors demonstrate that RowHammer attacks are completely viable under the assumptions we have described. And we will integrate the above points into the Threat Model section in our paper.
>
> [1] Kishani, Mostafa, Amirali Baniasadi, and Hossein Pedram. "Using silent writes in low-power traffic-aware ECC." International Workshop on Power and Timing Modeling, Optimization and Simulation. Berlin, Heidelberg: Springer Berlin Heidelberg, 2011.
>
> [2] Rakin, Adnan Siraj, Zhezhi He, and Deliang Fan. "Bit-flip attack: Crushing neural network with progressive bit search." Proceedings of the IEEE/CVF International Conference on Computer Vision. 2019.
>
> [3] Yao, Fan, Adnan Siraj Rakin, and Deliang Fan. "{DeepHammer}: Depleting the intelligence of deep neural networks through targeted chain of bit flips." 29th USENIX Security Symposium (USENIX Security). 2020.
>
> [4] Coalson, Zachary, et al. "Prisonbreak: Jailbreaking large language models with fewer than twenty-five targeted bit-flips." arXiv preprint arXiv:2412.07192 (2024).
>
> [5] Das, Sanjay, et al. "GenBFA: An Evolutionary Optimization Approach to Bit-Flip Attacks on LLMs." arXiv preprint arXiv:2411.13757 (2024).
>
> [6] Laplante, Phillip A. Comprehensive Dictionary of Electrical Engineering. Springer, 1999, p. 540. Entry: "refresh cycle". ISBN 3540648356.
>
> [7] Intel Corporation, Intel 64 and IA-32 Architectures Software Developer’s Manual, Vol. 3: System Programming Guide, April 2024.
>
> [8] Kim, Yoongu, et al. "Flipping bits in memory without accessing them: An experimental study of DRAM disturbance errors." ACM SIGARCH Computer Architecture News 42.3 (2014): 361-372.
>
> [9] Gruss, et al. "Rowhammer. js: A remote software-induced fault attack in javascript." International conference on detection of intrusions and malware, and vulnerability assessment. Cham: Springer International Publishing, 2016.
>
> [10] Tatar, Andrei, et al. "Throwhammer: Rowhammer attacks over the network and defenses." 2018 USENIX Annual Technical Conference (USENIX ATC 18). 2018.
>
> [11] Van Der Veen, Victor, et al. "Drammer: Deterministic rowhammer attacks on mobile platforms." Proceedings of the 2016 ACM SIGSAC conference on computer and communications security. 2016.
>
> [12] Mutlu, et al. "Rowhammer: A retrospective." IEEE Transactions on Computer-Aided Design of Integrated Circuits and Systems 39.8 (2019): 1555-1571.
>
> > **W2&Q2: Optimization Strategy Confusion and Generalizability About the Bit-selection Strategy**
>
> In our attack framework, identifying and flipping vulnerable parameters is the most critical step. In each module, we rank the parameters based on the absolute value of their gradients and select the top-k parameters with the highest gradient magnitudes as vulnerable bits. And for each selected parameter, flipping its most‑significant bit (MSB)—the bit whose inversion produces the largest numerical perturbation—tends to induce the greatest decrease in the overall loss.
>
> For different quantization formats, the MSB may vary. The table below shows the binary representations for three different 4-bit quantization formats.
>
> | Binary | INT4|   FP4| NF4|
> |:-------------:|:---------:|:---------:|:---------:|
> | 0000   | 0   | 0  | -1  |
> | 0001   | 1  | 0.0625  | -0.6962 |
> | 0010   | 2  | 8  | -0.5251 |
> | 0011   | 3  | 12 | -0.3949 |
> | 0100   | 4  | 4  | -0.2844 |
> | 0101   | 5  | 6  | -0.1848 |
> | 0110  | 6  | 2  | -0.0911 |
> | 0111  | 7  | 3  | 0  |
> | 1000  | -8 | -0    | 0.0796  |
> | 1001  | -7 | -0.0625 | 0.1609  |
> | 1010  | -6 | -8 | 0.2461  |
> | 1011  | -5 | -12 | 0.3379  |
> | 1100  | -4 | -4| 0.4407  |
> | 1101  | -3 | -6 | 0.5626  |
> | 1110  | -2  | -2  | 0.7230  |
> | 1111  | -1 | -3   | 1 |
>
> For INT4 quantization, since weights are stored in binary two’s complement in computers, flipping MSB—which is the sign bit—consistently results in the largest magnitude change. The same principle applies to INT8.
>
> In contrast, for FP4, due to its non-uniform quantization characteristics, flipping the sign bit does not consistently yield the largest magnitude change. For instance, flipping `'0000'` to `'1000'` results in zero change, whereas flipping `'0011'` to `'1011'` results in a change of 24. In such cases, if a vulnerable parameter is encoded as `'0000'`, flipping the sign bit has no effect on the model, which weakens the attack effectiveness under the FP4 format.
>
> To address this, we propose a specific-LUT strategy that identifies, for each parameter value, the bit whose flip leads to the maximum magnitude change. For example, for `'0000'`, flipping the second bit from the right yields `'0010'`, corresponding to a value of 8, which is the maximum possible change in that case.
>
> For NF4 quantization, although it is not based on two’s complement representation, we observe that flipping the sign bit still results in the maximum change. For instance, flipping `'0000'` to `'1000'` results in a magnitude change of 1, while flipping `'0011'` to `'1011'` results in a smaller change of 0.7328.
>
> GPTQ and AWQ are among the most widely used quantization methods today. The default `Auto-GPTQ` and `Auto-AWQ` libraries currently support only integer-based formats, such as INT4 and INT8. Therefore, LUT-based strategy is not supported for these methods. Among the mainstream quantization formats, only FP4 requires a LUT-based bit selection strategy.
>
> Although the LUT-based strategy cannot be applied to other mainstream quantization formats, our attack method remains effective across different quantization formats and quantization schemes. To validate this, we conduct additional experiments on INT4-quantized models using GPTQ and AWQ, and NF4-quantized models using the `Bitsandbytes` library.
>
> We select the **LLaMA-3.1-8B-Instruct** model as the target, and follow the same experimental setup as in the main paper: flipping 100 selected bits and evaluating the model on the DROP dataset. We evaluate the impact of the attack in terms of performance degradation, output fluency, and perplexity changes.
>
> |   Format | Accuracy (Before/After Attack) ↓ | Naturalness (Before/After Attack) ↑ | PPL (Before/After Attack) ↓ |
> |:----------:|:---------------------------:|:-----------------------------------:|:----------------------------:|
> | GPTQ-INT4  | 46.2 / 0.7  | 84.7 / 59.2    | 18.6 / 133.7    |
> | AWQ-INT4   | 45.8 / 1.9  | 79.4 / 57.5  | 20.1 / 146.4       |
> | NF4 | 47.9 / 0.0  | 83.8 / 53.8 | 19.7 / 154.3    |
>
> Results in the table show that the attack is effective across all quantization formats, and that the quantization type does not significantly affect attack success, demonstrating the strong generalizability of our attack across diverse quantization schemes.
>
> In future work, we will further explore additional quantization formats and their corresponding bit selection strategies.

---

> > ### Comment · Reviewer_y1pY · 2025-08-01
> >
> > Dear Authors,
> >
> > Thanks for your clarifications. I have carefully read your responses. Honestly, I like this work, as it has clear motivation and a well-structured methodology, without overly fancy decoration of the method. I will support this work for acceptance. Thanks for your work, and I hope your work can be open-sourced to the public for contributing to the community of $\textbf{Adversarial Machine Learning}$.
> >
> > Best Regards.
> >
> > The Reviewer

---

> > > ### Author Response · Authors · 2025-08-01
> > >
> > > Thank you very much for your kind and encouraging feedback. We truly appreciate your recognition of our work’s motivation, clarity, and methodological structure. Your support for acceptance means a great deal to us.
> > >
> > > We also share your belief in the importance of open research. We are actively working towards releasing the code and will make it publicly available upon final acceptance.
> > >
> > > Once again, thank you for your thoughtful review and support.

---

### Note · Authors · 2025-08-14

Dear AC and all reviewers,

We sincerely thank all reviewers for their careful reading and constructive feedback during the rebuttal process.

- We are especially grateful to **Reviewer y1pY** for the kind appreciation of our work, explicitly supporting its acceptance and praising its clear motivation and well-structured methodology.

- **Reviewer 4g7q** found our responses comprehensive, affirmed that the transferability concerns were effectively addressed, and recognized the work’s contribution and significance.

- **Reviewer NWPF** agreed that our revised caption improves Figure 2’s readability, was convinced of the approach’s practical feasibility (including white-box access and robustness considerations), and we will highlight our advantages over prior work more explicitly in the final paper.

- We also provided a detailed response to **Reviewer Vbmf**’s concerns, included additional results on SilentStriker’s performance against defense methods, and added references on BFA targeting general DNNs.

To conclude, we hope these updates address all concerns and further establish SilentStriker as a valuable contribution to the community.

Sincerely,

The Authors

---

### Decision · Program_Chairs · 2025-09-17

**Decision:**

Accept (poster)

**Comment:**

This paper proposes SlientStriker, a new bit-flip attack against LLM inference. The threat model targets edge device deployment, where the attacker has white-box access to the model architecture and parameters, and can manipulate a small number of bits in the memory addresses that host the model parameters to cause performance degradation. The unique aspect of SilentStriker compared to prior work on bit-flip attacks is its stealthiness, preserving the LLM's output naturalness.

Reviewers raised a few concerns during the discussion, including:
1. Confusion about threat model. What are the attacker's capabilities and why is this realistic?
2. Limited practical importance. What incentive does the attacker have to execute this attack in practice? Why is stealthiness crucial for the attack's success?

The authors mostly addressed concern 1 in the rebuttal, but the response to concern 2 is not very convincing. One way to make the attack scenario more convincing is to tie it to attacker goals in prompt injection. For example, in PI attacks, the attacker may embed instructions such as "ignore your previous instruction" and cause the LLM to call a harmful tool, e.g. send the user's API key to the attacker's server. The attacker has a strong economic incentive to execute this form of *targeted attack*, instead of just deteriorating model performance via an *untargeted attack*. Similarly, for bit-flipping, an attacker that can force the model to call harmful tools in a stealthy manner would represent a much more realistic attack goal than deteriorating model performance.

The authors are strongly encouraged to reconsider their motivation for studying this attack to align it to a more realistic attacker goal. With this caveat in mind, AC recommends acceptance to NeurIPS.